# A qualitative process evaluation within a clinical trial that used healthcare technologies for children with asthma–insights and implications

Louisa Lawrie[1]*, Stephen Turner[2], Seonaidh C. Cotton[1], Jessica Wood[1], Heather M. Morgan[3]

1 Health Services Research Unit, Institute of Applied Health Sciences, School of Medicine, Medical Sciences and Nutrition, University of Aberdeen, Aberdeen, United Kingdom, 2 Royal Aberdeen Children's Hospital, University of Aberdeen, Aberdeen, United Kingdom, 3 Institute of Applied Health Sciences, School of Medicine, Medical Sciences and Nutrition, University of Aberdeen, Aberdeen, United Kingdom

* louisa.lawrie1@abdn.ac.uk

**Data Availability Statement:** All relevant data are within the manuscript and its Supporting Information files.

## Abstract

### Background

Healthcare technologies are becoming more commonplace, however clinical and patient perspectives regarding the use of technology in the management of childhood asthma have yet to be investigated. Within a clinical trial of asthma management in children, we conducted a qualitative process evaluation that provided insights into the experiences and perspectives of healthcare staff and families on (i) the use of smart inhalers to monitor medication adherence and (ii) the use of algorithm generated treatment recommendations.

### Methods

We interviewed trial staff ($n = 15$) and families ($n = 6$) who were involved in the trial to gauge perspectives around the use of smart inhalers to monitor adherence and the algorithm to guide clinical decision making.

### Findings

Staff and families indicated that there were technical issues associated with the smart inhalers. While staff suggested that the smart inhalers were good for monitoring adherence and enabling communication regarding medication use, parents and children indicated that smart inhaler use increased motivation to adhere to medication and provided the patient (child) with a sense of responsibility for the management of their asthma. Staff were open-minded about the use of the algorithm to guide treatment recommendations, but some were not familiar with its' use in clinical care. There were some concerns expressed regarding treatment step-down decisions generated by the algorithm, and some staff highlighted the importance of using clinical judgement. Families perceived the algorithm to be a useful technology, but indicated that they felt comforted by the clinicians' own judgements.

**Funding:** The study was supported by a grant awarded by the National Institute for Health Research (Efficacy and Mechanism Evaluation programme; reference 15–18–14). Circassia Ltd supplied 16 NIOX VERO® machines and associated consumables in support of the study. The funders had no role in study design, data collection and analysis, decision to publish, or preparation of the manuscript.

**Competing interests:** The authors have declared that no competing interests exist.

## Conclusion

The use of technology and individual data within appointments was considered useful to both staff and families: closer monitoring and the educational impacts were especially highlighted. Utilising an algorithm was broadly acceptable, with caveats around clinicians using the recommendations as a guide only and wariness around extreme step-ups/downs considering contextual factors not taken into account.

## Introduction

Technological advancements in healthcare are rapidly evolving and transforming patient experiences and outcomes both within the UK and around the world [1–4]. Several national and global policies and frameworks have been developed to encourage and govern the introduction of healthcare technologies into standard practice [1, 5, 6]. The advantages of healthcare technology over "traditional" healthcare include standardising care, limiting medical errors, providing better coordinated care, improving effectiveness and cost effectiveness of interventions, equity and increasingly pro-active surveillance [7]. Healthcare technologies can be used to monitor and manage patient results, communicate health related information, allow real-time outcomes to be measured and facilitate clinical decision making in a range of chronic conditions [7–9], including asthma [10–12].

Childhood asthma care is an area in which technology is starting to be introduced [13]. Over 8 million people in the UK have been diagnosed with asthma, and the incidence is higher in children than in adults [14]. The prevalence of asthma in the paediatric population (worldwide) is approximately 11% [14]. Healthcare technologies could potentially help many individuals manage their asthma. One form of healthcare technology which is increasingly used as part of asthma management is the smart inhaler, or adherence monitor, and this provides an objective measure of adherence to asthma preventer treatment, delivered by inhaler [15]. A second genre of healthcare technology used for the provision of care of childhood asthma is digital support tools, or algorithms, which aid clinical decision making [15, 16].

Parent and healthcare staff perspectives on healthcare technologies, such as smart inhalers, for asthma care in children have been described in prior studies [17, 18]. The use of algorithms in decision making for asthma care has been explored [19], but not specifically within the context of childhood asthma. Within a randomised clinical trial of asthma management in children, we conducted a qualitative process evaluation which provided insight with regards to the experiences and perspectives of healthcare staff and families on (i) the use of smart inhalers to monitor medication adherence and (ii) the use of algorithm generated treatment recommendations.

## Methods

### Context

The current study was embedded within the RAACENO trial (Reducing Asthma Attacks in Children using Exhaled Nitric Oxide) [20] (trial registration: ISRCTN67875351). RAACENO explored whether the addition of exhaled nitric oxide measurements to usual care reduced asthma attacks among children, and used both smart inhalers and a web-based algorithm to inform treatment decision making. The trial involved 12 months of participation by 509 children aged 6–15 years, with assessments 3, 6, 9 and 12 months after randomisation. Participants were randomised to the intervention group, which used Fractional Exhaled Nitric Oxide

(FeNO) measurements to inform treatment decisions, or the control group which did not consider FeNO measurements. In both groups, treatment recommendations were guided by a web-based algorithm developed for the purpose of this trial (described elsewhere–[21]). The algorithm considered current asthma symptoms, treatment adherence, current treatment and (in the intervention arm) FeNO measurements. The research teams entered data into the algorithm at baseline and at each follow-up visit, and this algorithm made a recommendation in terms of stepping up (increasing) the child's asthma preventer treatment, stepping down (decreasing) the treatment, or maintaining the same treatment. Participants in both groups were provided with a smart inhaler to measure adherence to treatment.

## Design

This was a primary qualitative process evaluation that used semi-structured interviews with trial staff and families (children with their parents). The process evaluation initially set out to explore experiences and ascertain acceptability of the trial intervention, and to solicit in depth feedback on taking part in the trial from the perspectives of both families and trial staff. The process evaluation used qualitative interviews to collect data, but it was not designed to be an in-depth qualitative study with an underpinning philosophy used to guide the conduct of it. We nevertheless elected to report the findings in this paper as additional insights that emerged around the use of technology in the management of childhood asthma.

Two separate topic guides were used for interviews with trial staff and families (see S1 Appendix). These were designed and developed by two researchers (HMM and DB). The topic guides were revised after a short internal pilot. Audio recordings of the practice interviews were assessed by two researchers (LL and HMM) who adjusted the order of the questions presented within both topic guides based on the content of the audio recordings and in response to feedback received from the (pilot) interviewees. An iterative approach was adopted to revising the topic guides throughout the qualitative process evaluation [22]. Both interview guides explored perspectives related to recruitment, randomisation, the smart inhalers, algorithm, and the future of asthma management.

## Sampling and recruitment

**Staff.** We aimed to recruit five research nurses, based on the literature and our experience of conducting qualitative health research. We revised this target to 15–20 participants (expanded to include different roles)–this was in response to informal feedback and conversations with sites. We elected to recruit more staff members compared to families because they played a key role in implementing the algorithm recommendations. We also considered that there was a diversity and richness of experiences and insights to be gleaned from staff at different sites and performing diverse roles.

Email invites were distributed to a purposive sample of members within research teams who occupied various roles in the trial (e.g., Research Nurses, Consultants and Principal Investigators) across ten different UK sites. These sites were selected to obtain interview data from staff who worked with different levels of recruitment to the RAACENO trial (smaller vs. larger numbers) and based on trends that demonstrated the sites associated with low/high adherence to algorithm recommendations. However, due to the COVID-19 pandemic and the resultant low uptake of individuals responding to our initial invite at the time, we elected to adopt a convenience sampling method to improve recruitment figures. Reminder invites were subsequently distributed and staff at three additional sites were approached. Interviews with trial staff who agreed to participate were arranged and conducted by a researcher (LL) who was not known to the participants prior to data collection.

Fifteen trial staff members were included in the final sample size. An additional three individuals had initially expressed interest in participating but 1 was unavailable due to the pressures of the pandemic and 2 did not reply to further contacts. Data saturation became evident following the first 6 interviews [23], but data collection continued to ensure we obtained a more representative sample of staff occupying various roles within the trial and at different sites [22, 24]. The researcher introduced herself as a qualitative researcher with no involvement in clinical data collection. Interviews were conducted via telephone, audio-recorded and transcribed by an external transcription company. Field notes were generated to facilitate data analysis. Participants were interviewed from March 2020 till June 2020.

**Families.** We set out to interview 20 families drawn from both arms of the trial. Families were first approached and invited by research nurses across seven trial sites. Site selection was initially based on recruitment levels at sites and the convenience of accessing the hospitals for face-to-face interviews based on geographical location (prior to the requirement for remote working due to COVID-19). Thereafter, potential interviewees identified by research nurses as interested in being interviewed were followed up by one researcher (HMM) by telephone during the study and then approached again by the research nurses at their 6-, 9- or 12-month assessments to ask whether they would still be happy to be interviewed following their 9-month (penultimate) or 12-month (final) assessment (see S1 Appendix for further information regarding participant follow-up procedures). Of the 17 families who initially expressed interest in participating, 6 families (mother-child pairs) were interviewed between March 2020 till May 2020. The remainder of those who initially expressed an interest in taking part did not reply to further invitations. Interviews were conducted by a researcher (LL) who had no previous correspondence with the families and who introduced herself as a qualitative researcher independent from the clinical team involved in the data collection and analysis for the host trial. Interviews were conducted via telephone, audio-recorded and transcribed by an external transcription company. Field notes were generated to facilitate data analysis. Recruitment continued for both trial staff and families until data saturation was reached.

## Ethics

This RAACENO study, including this interview component, was reviewed and approved by the North of Scotland Research Ethics Committee (IRAS 212541). Informed consent was obtained from all participants. Written consent was provided by 3 staff members who were interviewed at the beginning of the data collection process. Due to the pandemic, verbal consent was obtained (and approved by the ethics committee) for the remainder of the interviews. Verbal consent was audio-recorded and transcribed verbatim (provided by parents for the family interviews).

## Data analysis

A thematic approach was used to analyse transcripts from the qualitative interviews within and across cases, and data management was assisted by Microsoft Excel/Word. Initially, one researcher (LL) identified key themes and categories by listening to the interviews via the audio recordings and reading the associated transcripts. This was part of the process of coding data and developing themes. A second researcher (HMM) conducted a more granular analysis, independently, by reviewing the transcripts line-by-line and generating common themes representing topical ideas that were identified within the data. Both researchers (LL and HMM) met to discuss the themes that resulted from these independent analyses and agreed on the key findings from within the data. Notes taken during interviews by the interviewer (LL) were referred to during analysis. Transcripts were not returned to participants for feedback. Findings were reviewed by the local RAACENO team and trial steering committee.

**Table 1. Characteristics of our staff sample.**

| Site ID | No. interviewed* | Roles | Staff identifiers** |
|---|---|---|---|
| 1 | n = 2 | • Research Nurse<br>• Consultant Paediatrician | 1CT1, 1RN2 |
| 2 | n = 2 | • Research Nurses | 2RN1, 2RN2 |
| 3 | n = 1 | • Consultant Paediatrician | 3CT1 |
| 4 | n = 2 | • Research Nurse<br>• Consultant Paediatrician | 4RN1, 4CT2 |
| 5 | n = 2 | • Research Nurses | 5RN1, 5RN2 |
| 6 | n = 3 | • Research Nurses (n = 2)<br>• Consultant Paediatrician | 6RN1, 6CT2, 6RN3 |
| 7 | n = 1 | • Research Nurse | 7RN1 |
| 8 | n = 1 | • Research Nurse | 8RN1 |
| 9 | n = 1 | • Research Nurse | 9RN1 |

* Where more than one staff member was interviewed per site, each was interviewed independently

** e.g., Site number (1), Role (CT), with the second number denoting the order in which the interviews were carried out at a given site

The reporting of this qualitative process evaluation complies with the COREQ checklist [25]. Where direct quotes are presented in the findings, we have used unique identifiers to attribute data to the individuals who took part in the qualitative process evaluation, while preserving their anonymity.

## Findings

### Staff

Interviews with trial staff members lasted between 18 and 34 minutes.
Characteristics of participants are provided in Table 1.

### Families

Children's ages ranged from 12–16 years. These interviews lasted between 28 and 45 minutes. All families (mother-child pairs) who participated had been randomised to the intervention arm of the trial. After four interviews, no new themes emerged. Subsequent interviews confirmed these themes, and it was agreed that data saturation had been reached [24].

### Data

There were some differences between family and staff perspectives on the smart inhaler and algorithm. Findings present the views of staff members, prior to outlining family perspectives. Table 2 highlights the dominant themes of the interviews, resultant from the data analysis.

**Table 2. Key themes arising from perspectives of trial staff and families regarding the smart inhaler and algorithm.**

| Topic | Interviewees | |
|---|---|---|
| | *Staff* | *Parents & Children* |
| Smart inhaler | Technological challenges;<br>Monitoring adherence;<br>Enabling communication. | Technological challenges;<br>Motivation to comply and accountability for medication adherence. |
| Algorithm | Familiarity;<br>Concerns regarding step down recommendations;<br>Using clinical judgement | Perspectives on usefulness;<br>Enabling communication;<br>Comfort in clinician judgements. |

## Staff

**Smart inhalers.**    *Technological challenges*. Overall, smart inhalers were perceived to be generally appreciated, but with some technical problems relating to recording/uploading data (9RN1, 8RN1, 7RN1, 6CT2, 5RN1, 4RN1, 3CT1). Accuracy of the smart inhaler data indicating patient usage was questioned by one staff interviewee (3CT1). There was disappointment if a score had to be disregarded:

"In the beginning a few seemed to work, but towards the end I was disappointed not to be able to show the parents good results if they said their child was being compliant. . .So we had to you know, to ignore the [adherence] score even though the child. . . the child wanted to beat their last score and it could be disappointing. . ." (2RN1).

*Monitoring adherence*. Smart inhalers were considered good for monitoring adherence (8RN1) and providing an objective index of adherence (6RN1), as well as for setting targets:

". . . The ones that were downloaded, they were really happy with them and it gave them a focus. One of my early recruits, hers always worked amazing and her mum was really good at encouraging her and saying, "Right, we want the number higher next time, come on, we want better there," you know? So really used it as a tool to encourage them. . ." (5RN1)

Staff believed that adherence improved (6CT2) and thought that smart inhalers for non-adherent children (9RN1) were 'a bit of a game' with tech to monitor adherence (7RN1) and to manage some patients (6RN3), although it was acknowledged that some would probably not take their inhalers properly even if monitored (6RN3). For example:

"What you don't want to be doing is to continually escalate doses down an algorithm when in fact the reality is, is they're not taking their medicine at the required frequency." (4CT2)

Some families were reportedly excited about seeing the results and their performance (6RN1), but it was identified as a potential reason for drop out of children who were not adherent (7RN1). However, even when adherence was low, families did not seem to mind being monitored (6CT2), although one clinician mentioned the 'big brother' factor:

"I think there is a big brother factor, they get a bit anxious about us scrutinising how well they take their medicines. But having something that brings it to the forefront so that you can have an honest discussion I think is very, very valuable." (4CT2)

*Enabling communication*. The data were useful for facilitating 'difficult conversations', for example, helping to have conversations and improve adherence to see asthma under better control (3CT1); discussing when adherence had drifted off, e.g., during holidays (8RN1); and uncovering when a child had not been taking medication with parents assuming that they were (7RN1). Improved adherence of some was reported initially, but this was not always sustained (6RN3, 4CT2), although the smart inhalers could be used as an incentive to obtain a score (5RN1, 4CT2, 2RN1). For example:

"I think the children enjoyed that game you know, they always tried to guess what it was going to be and then they tried to beat their score the next time, and they understood how to achieve that and what the significance that inflammation can mean in combination with

their lung function test. It helped them to understand it a little bit better and the importance of using their inhaler as per what the doctor said." (2RN1)

Staff perceived that parents wanted access to their child's data so that they could monitor them (8RN1) and suggested that it would be better if there was an opportunity for instant download at home rather than in clinic (2RN1). Factors that were concerning staff around data capture, also highlighted above in terms of missing data affecting the algorithm's recommendation, were: weekends when children go to the other parent and forget the smart device (6CT2, 2RN1) and cut offs/timings automated within the device:

"... So, if you've got a teenager that's getting up at 1.00 in the afternoon, they'll get up, they'll take their inhaler and then they'll go and do whatever, and then they'll take it again, but that was saying "No" for the morning dose because it had a cut off of 12.00..." (6CT2)

Otherwise, positive experiences were reported:

"I think it's quite nifty, it's not heavy, it's not bulky, it's quite easy. So, I think yeah, it's not too bad to use, it's quite easy, yeah." (6CT2)

And being able to check on other aspects was also considered as helpful, e.g., expiry date (8RN1).

## Algorithm

**Familiarity with algorithms—staff perspectives.** There were a range of considerations regarding experiences and acceptability of the algorithm's role in the diagnostic and decision-making process. Familiarity with using algorithms in treatment was a factor: one staff member said that they had never used algorithms before to determine treatment (4RN1). Another expressed a very clear interpretation of what should happen, suggesting that a 'purist' would follow the algorithm to the letter (9RN1). Staff seemed to express an open mind to using such an approach in clinical practice. But there were some caveats, for example, as long as it was considered 'reasonable' (8RN1) or 'appropriate' (6CT2), staff said they would accept the recommendation, however, if not, staff explained how they would use it:

"If you were stepping down as you went into the school return in September it made you more anxious if you knew that in historical years that they'd had asthma attacks at that time. So that did influence a minority of patients in terms of whether the algorithm was followed or not." (4CT2)

*Concerns regarding step down recommendations.* There was concern if a step-down was recommended in the context of continuing emergency department visits and needing to look more closely (1CT1) or causing emergency department visits through a step-down that led to exacerbation (6CT2). The use of the algorithm was considered to be beneficial when a step down meant that a patient could be returned to community care rather than being managed in secondary care (7RN1). For this participant, the algorithm recommendations provided additional evidence which harmonised with their clinical judgement. Internal 'missing data' problems and external contextual factors were also considered, for example one staff interviewee said that there may be a problem if the algorithm was not operating on a full dataset due to missing data (6RN1), also highlighted elsewhere where data were missing because the child was staying with their other parent and forgetting their smart inhaler (6CT2).

*Using clinical judgement.* There were perceptions that people–both staff and families–were not accepting of 'technology' on its own:

> "I think most families who were accepting of the algorithm but only with the proviso that it's never just the technology and the algorithm, and the doctor can always override that if necessary." (2RN2)

> "Reassuring them that the computer's not dictating to them what's going to happen, that the doctor will overall decide what's safe for them." (2RN1)

Clinician judgement was considered to be critical to applying the algorithm's recommendation (6RN1) and the notion that understanding among trial participants that the algorithm recommends, and the clinician decides, was highlighted (7RN1). Trust in the consultant and families accepting clinical judgement, whether or not this involved adhering with the algorithm (9RN1), was raised. The notion of 'computer says no' and that algorithms are pretty limited was raised, alongside the idea that patients and their families come to see an expert in whom they trust and with whom they have a personal relationship, which cannot be replaced by a computer and algorithm (3CT1). Staff perceived that parents were sometimes reluctant to go 'too low' or alarmed about 'how high' (8RN1) the algorithm-recommended treatment, and if a parent was unhappy about a step down, staff believed that their feelings needed to be taken into account (6CT2). One staff member indicated:

> "Yes, actually fairly frequently, we haven't agreed with the algorithm. Well, for various reasons. Probably more often that parents have felt a bit more cautious when a step–down is suggested." (2RN1)

Research nurses checking adherence with the algorithm rather than parents was noted by one site (1CT1) and not using the word 'algorithm' with families was highlighted at another (7RN1). Ultimately, providing the best care was the main factor:

> "To them it was just like, at the end of the day, we're going to give you the best care we can and the best treatment" (5RN1)

In terms of rates of agreement with algorithm recommendations, sites indicated that they had different experiences. For example, at one site, there was an estimate of disagreeing about 20% of the time due to the time of year and triggers (apprehensive about stepping down, 5RN1), whereas at another site, it was suggested that there was disagreement about half of the time (4RN1). One site perceived that the algorithm was 'chopping and changing' and preferred more settled treatment (9RN1), whereas another site considered that there were 'too big jumps' (8RN1). However, sites expressed satisfaction around obtaining data:

> "One of our consultants, in particular, struggled with that a little bit, because you could have a child that would have. . . so step–downs were frequently suggested and when actually their lung function was decreasing, with those patients we had to override the algorithm." (2RN1)

The need for GPs to have a better tool for use in primary care, i.e., using the algorithm, was noted (1CT1).

## Families

**Smart inhalers.** *Technological challenges*. Smart inhalers were considered to be useful, but many encountered problems with their accuracy (Family 1, Family 2) and reported that the battery required regular charging (Family 4, Family 3). One parent said that it was difficult to use, even for adults:

". . .Obviously from my experience, I work as support worker and when I give my residents the smart inhaler, they don't know what to do with it, so some. . . even adult people don't know what to do with this. . ." (Family 1)

There was a suggestion of a battery indicator being helpful (Family 5) and also that children are more tech savvy and would manage charging better than their parents (Family 4). One parent suggested having the option to download at home, perhaps weekly, to monitor their child would have been useful (Family 3). Another parent wanted to buy a smart inhaler:

". . .As a parent, it's been unquestionably my favourite thing about the study. I really echo what she said, I wish that we could buy it. I mean, I said I would buy it for somebody else in the study that maybe couldn't afford it, I have no idea how much they are, but I know they said it would be difficult to get the software or something on a phone. . ." (Family 6)

*Motivation and accountability for medication adherence*. Families (parents/children) reported that they liked being monitored (Family 4) and the computer being able to account for adherence to generate recommendations after technical issues were fixed. Most families specifically mentioned that the monitoring was a motivator to be adherent with treatment, even when the asthma was well managed (Family 5, Family 4, Family 2, Family 3, Family 6). One parent said:

". . . in my opinion it makes the child a little bit more conscious about making sure they take it knowing that it's being recorded. . ." (Family 4)

Two parents described that a child feeling more responsible was a positive consequence of smart inhaler usage (Family 5, Family 6). Smart inhalers encouraged patients to adopt a degree of accountability in terms of their medication adherence. One parent said that their child had learned to manage their asthma better due to their study participation (Family 5).

**Algorithm.** *Perspectives on usefulness*. One parent felt that the algorithm made good decisions (Family 5), and another mostly agreed with both the step-ups and step-downs in preventer treatment (Family 4): this was linked to their perception about how the condition was experienced by their child at the time. Another liked the close monitoring with recommendations about stepping up and down and 'being on the right amount':

"No, the doctor always just did what the computer recommended. But he always explained it and always said, you know, he always kind of justified it because her asthma had been fine, or because, you know, she hadn't been on steroids for the previous three months or whatever. So, he always talked it through." (Family 3)

The fluctuating nature of asthma symptoms (Family 4, Family 3, Family 6) were identified by parents as being relevant to how the algorithm was used. It was noted that the use of the algorithm required flexibility based on the family/patient lifestyle. For example, one family did not adhere to the treatment recommendations for a month, after consulting a clinician,

because of travel plans to an area of poor air quality (Family 3). One parent cited that they respected decisions based on the algorithm because they trusted the clinical staff:

> "It's a programme that works out information. I don't know anything about that, very technology behind, but yeah, they did explain it to me. I just trusted that they knew what they were doing." (Family 2)

Some parents remembered the word 'algorithm' (Family 5) and one said that it was communication on multiple occasions (Family 4), whereas others said they did not remember the word, but felt that the language used by staff to describe the trial interventions was appropriate to their level of understanding (Family 3). In response to questions designed to elicit perceptions of the role of the algorithm in treatment decisions, one said it was initially daunting, but that this seemed acceptable for making recommendations about treatment and was described as 'clever':

> "...the computer seemed to pick up on all of that, which I think was...I think it was really, really clever. But obviously having... knowing that there was a doctor overseeing it as well was that little bit of reassurance." (Family 4)

Most parents reported that the algorithm had suggested changes, a step-up/down from current treatment recommendations, on one to three occasions (Family 1). Some liked the fact that a step-down was recommended as this meant taking less medication (or had been taking more than required) (Family 1), but others did not and were anxious about a step-down, even though they also wanted less steroid (Family 6), or a maintained level of treatment based on discussion between parents and clinicians, even where the algorithm suggested no treatment (Family 5). One parent considered algorithm recommendations to be reliable as they seemed to recommend a step-up when the asthma had felt bad but suggested that it would have never recommended no treatment (no medication) and as such was reliable:

> "... She did have a bad asthma year the year of the study, but not particularly worse than it would have been anyway, I don't think. She's just got pretty bad asthma. But I liked that they were very responsive to everything." (Family 3)

*Enabling communication.* Regarding a step-up decision, one parent liked to be able to have a conversation about the computer recommendation and was reassured that their clinician would not follow the algorithm if everyone perceived a step-up was too big a step (Family 4). Another felt similarly confident that, should they want to override it, for example because the child was not keen due to increased usage of steroids [and the perceived consequence of reduced growth], their consultant would step in:

> "... she was really paranoid about the more steroids that she took, the shorter she would end up. So, every time the computer said she needed to increase her steroid dose she said, 'Oh I don't think so'." (Family 2)

*Comfort in clinician judgements.* The majority of trial participants' parents reported feeling comforted by the idea that they could ultimately depend on a doctor's interpretation and overriding the algorithm's recommendation based on knowledge of the case and adapting to season (Family 1, Family 5), or taking into account a previous bad reaction (Family 6). The fact that the algorithm was being used as a guide and not on its own (Family 5, Family 4, Family 6) was identified as important in the acceptability of the algorithm.

"...I'm fine with technology helping advise the doctors, but ultimately the doctor made the call, using the technology and the finding." (Family 6)

## Discussion

### Principal findings

Smart inhalers were welcomed by families and clinicians, especially regarding adherence [26, 27]. Staff welcomed data as a way to raise the subject of adherence, a known contentious issue [28] especially with children [29]. Smart inhalers were not considered as user friendly by our interviewees compared to a previous study [30]. Staff, but not parents, had concerns about 'big brother' and surveillance, despite the perceived clinical benefit (see research note [31] which discusses interpretations of surveillance and digital technology in healthcare).

For the algorithm, both staff and parents (and children) expressed confidence in the recommendations, but also maintained how having clinical oversight was important. This is in line with discussions in other areas of healthcare about the role of algorithms and the development of artificial intelligence to guide decision making [32]. Lennartz et al [33] described a sense of "cautious optimism" amongst patients who perceived the clinical capabilities of artificial intelligence in a positive light, yet regarded physician input as superior.

Concerns around strictly following the algorithm's treatment recommendations were raised due to contextual factors not being taken into consideration beyond the data considered by the algorithm. Some technical issues/downsides with both smart inhalers (e.g. missing data) and algorithm (e.g. not accounting for seasonal changes) were reported, but the benefits of using novel technology in managing and treating asthma were recognised, especially objective tests [34].

As Lupton indicated [35], establishing the effective and responsible delivery of digital health technologies and collection, protection and sharing of health data is highly complex. Infrastructure, ethical and social issues need to be considered. The training of artificial intelligence using algorithms and patient data also needs to be transparent and accountable to build trust [36].

### Implications

The use of healthcare technologies within healthcare is increasing, and it is important to consider perspectives of stakeholders, e.g. clinical staff and patients. Doing so may limit misunderstandings and concerns regarding technology use. There are currently evidence-based clinical guidelines on the management of asthma available to standardise practices [37]. These guidelines include information about Computerised Decision Support Systems and its' use in clinical decision making. As technology is becoming more commonplace, we need clinical guidelines to standardise practices. For example, guidelines should outline suitable algorithms to follow and information pertaining to the implementation of algorithms within the appropriate pathway of treatment (e.g., primary or specialised care). Further research is required to assess whether nurses and consultants might differ in their opinions of algorithm clinical use, particularly if it is to become standardised practice [36].

### Strengths, limitations, and future research

The main strengths of this qualitative process evaluation are that it was conducted by a team of qualitative researchers with backgrounds in a range of disciplines, undertaken over time in phases, including developing materials and collecting data, with two researchers

independently coding and analysing all data [22]. The fact that we were able to collect data during a pandemic was also a strength.

Nevertheless, our sample was adapted to include more staff members, but we were unable to achieve our planned recruitment among families. We initially elected to recruit more staff members due to their key role in implementing the interventions within the trial–namely the algorithm. It would have been interesting to interview clinicians regarding intentional non-adherence. However, we consider that saturation was achieved among both staff and the heterogeneous family samples [38] within six interviews for each group, which we considered to be early. This could be attributed to sample bias, especially among families due to having only managed to recruit participants who were from the intervention group. Furthermore, given that this was a sub-study within a clinical trial, and this paper reports on that process evaluation, it was not possible to resource the recruitment of a larger sample. Researchers should consider our findings and whether it is necessary to pursue small/in-depth or larger surveys on related research to develop knowledge on this topic further.

The implications of the pandemic encouraged us to adopt a convenience method of sampling, as opposed to purposive, due to time constraints of healthcare professionals. It may have been more insightful to investigate perspectives of trial staff with varied years of clinical experience. Furthermore, as we were unable to interview parents and children separately, it is unclear whether they may have provided different responses.

Potential areas for further exploration include the use of lung function scores in algorithm decisions and the involvement of patients and families within trial and intervention designs. It is likely that the involvement of patients and families could improve the acceptability of technological interventions in the future.

## Conclusion

In this qualitative process evaluation, trial staff representing several roles and across different sites were interviewed to understand the acceptability of the intervention from their points of view. Several families (mother and child pairs) were interviewed about their experiences and views, too. Overall, experiences within both groups were positive. Key was that the use of technology and individual data within clinical appointments was considered useful: closer monitoring and the educational impacts were especially highlighted. We also ascertained that using an algorithm was broadly acceptable, with caveats around clinicians using the recommendations as a guide (rather than being dictated by it) and wariness around extreme step-ups/downs considering contextual factors not taken into account by the algorithm.

## Supporting information

**S1 Appendix. Participant (families) follow-up data collection procedure.**
(DOCX)

## Acknowledgments

We thank Dan Brunsdon for his involvement in the development of the interview topic guides. We also thank the participants for dedicating their time to be interviewed and for staff at recruitment sites who helped identify families for this qualitative study. We are grateful to Andrea Fraser for secretarial and data coordination support and to Ruth Thomas for help and advice in developing the grant proposal. We thank the programming team at the Centre for Healthcare Randomised Trials for developing and maintaining the study website which incorporated the treatment algorithm.

## Author Contributions

**Conceptualization:** Louisa Lawrie, Stephen Turner, Seonaidh C. Cotton, Heather M. Morgan.

**Data curation:** Louisa Lawrie.

**Formal analysis:** Louisa Lawrie, Heather M. Morgan.

**Funding acquisition:** Seonaidh C. Cotton, Heather M. Morgan.

**Investigation:** Louisa Lawrie, Heather M. Morgan.

**Methodology:** Louisa Lawrie, Heather M. Morgan.

**Project administration:** Louisa Lawrie, Seonaidh C. Cotton, Jessica Wood, Heather M. Morgan.

**Supervision:** Stephen Turner, Seonaidh C. Cotton, Heather M. Morgan.

**Writing – original draft:** Louisa Lawrie, Heather M. Morgan.

**Writing – review & editing:** Louisa Lawrie, Stephen Turner, Seonaidh C. Cotton, Jessica Wood, Heather M. Morgan.

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
