## [Decision Letter · Decision Letter 0]

26 Aug 2022

PONE-D-22-10382An interview study within a clinical trial that used healthcare technologies for children with asthma – insights and implicationsPLOS ONE

Dear Dr. Lawrie,

Thank you for submitting your manuscript to PLOS ONE. After careful consideration, we feel that it has merit but does not fully meet PLOS ONE’s publication criteria as it currently stands. Therefore, we invite you to submit a revised version of the manuscript that addresses the points raised during the review process.

 Apologies for the prolonged review period, it was a mainly due to the unavailability of suitable reviewers who would provide relevant and unbiased commentary and feedback. Please address all reviewer comments, specifically those required further justification and/or context.  

We look forward to receiving your revised manuscript.

Kind regards,

Daswin De Silva

Academic Editor

PLOS ONE

Journal Requirements:

2. Please provide additional details regarding participant consent. In the ethics statement in the Methods and online submission information, please ensure that you have specified whether: 1) whether the ethics committee approved the verbal/oral consent procedure, 2) why written consent could not be obtained, and 3) how verbal/oral consent was recorded. If your study included minors, please state whether you obtained consent from parents or guardians in these cases. If the need for consent was waived by the ethics committee, please include this information.

3. Please ensure you have included the registration number for the clinical trial referenced in the manuscript.

Reviewers' comments:

Reviewer's Responses to Questions

**Comments to the Author**

1. Is the manuscript technically sound, and do the data support the conclusions?

Reviewer #1: Partly

Reviewer #2: Yes

2. Has the statistical analysis been performed appropriately and rigorously? 

Reviewer #1: No

Reviewer #2: I Don't Know

3. Have the authors made all data underlying the findings in their manuscript fully available?

Reviewer #1: Yes

Reviewer #2: Yes

4. Is the manuscript presented in an intelligible fashion and written in standard English?

Reviewer #1: Yes

Reviewer #2: Yes

5. Review Comments to the Author

Reviewer #1: The topic is relevant, interesting. However, Line 80- 82 describes previous research conducted on the topic. This indicates the sole novelty of this work being a focus on the use of algorithms in decision making, within the context of childhood asthma. if so, would it have been more worthwhile to gather insights from a larger sample on this topic?

Line 73: Additional details relevant to the cited prevalence rate would be useful such as location. Additional information on the global burden of asthma or at least, in the UK would be useful additions here as well.

Line 117: The study design is described as a process evaluation, while the title indicates this is a qualitative study. Further, the title as well as early sections which preceded the Methods section point towards a study focusing on the lived experiences of staff and families in this context. The methods section does not sufficiently describe the focus on lived experience, which is essentially the methodology underpinning this work. It also seems quite irrelevant to explore perspectives related to recruitment and randomisation in this context. Please consider revisiting the framing of the methods section, consider the methodological merits of phenomenology in this research context and either changing the title to indicate this was a process evaluation or reframing the methods section to indicate being more strongly guided by qualitative methodology.

Line 135-140 and Line 151-156 are repetitive. Please consider a creative way to present this section to address this.

Line 143 states "set out to interview 20 families". However, the final sample size has not been described.

Line 169: Identification of themes post listening to the audio recording is not a commonly used method. Perhaps use a reference to indicate it has been used by others. Also, it would be useful to explain why member checks were not conducted (e.g. lack of resourcing). Especially, as Line 176 notes findings were reviewed by the clinical teams. Additionally, this is not traditionally an approach undertaken to ensure the findings authentically represent the user experience, which is the purpose of member checks.

Line 191-193: This seems a poor rationale for the sample size of families recruited into the study, given the focus is on medication adherence, and therefore in particular the experiences of the users.

If there were 17 families who expressed an interest in participating, did the participant information sheet state that recruitment would be discontinued once saturation had been reached? Did the research team notify the remaining prospective participants that they would not be interviewed as data saturation had been reached? I feel these are important aspects to be addressed in the study design and description.

The results section is interesting and has been presented well.

Line 473-474: is this a novel findings or replicates the study findings referred to in text as (31)?

Line 478: The strength of this claim will benefit from additional references, if they can be sourced.

Line 505-506: This claim is not entirely accurate as the methods section noted the first researcher developed themes post listening to audio recordings and therefore could not have undertaken coding of the data.

I wondered if the data revealed user and staff experiences related to using this technology during a pandemic. It would be an interesting angle to explore if there is existing data on this within the interviews.

Please review the reference list to ensure formatting is consistent (e.g. journal name is presented in upper and lower case )

Overall, the findings are relevant, and interesting. However, I would urge the authors to consider the concerns noted above and the overall strength and generalizability of these findings given the small sample size of parents interviewed. Perhaps, it is worthwhile considering the collection of additional data from families who meet the same inclusion criteria and increasing the overall sample size of the research undertaken.

Reviewer #2: The authors evaluate healthcare technologies (including treatment recommendation algorithms and smart inhalers) using clinical trials in childhood asthma.

The paper is logically structured and well-written. The authors could expand the introduction including additional information on the prevalence of asthma.

In the method, it is recommended to include final sample sizes for staff and families and add justification for the sample size selections.

It is recommended to justify through references the use of audio recordings.

The results are presented well. The authors could strengthen the claims with additional references.

It would be interesting to expand the finding related to the use of technology in pandemics if applicable.

6. PLOS authors have the option to publish the peer review history of their article (what does this mean?). If published, this will include your full peer review and any attached files.

Reviewer #1: No

Reviewer #2: No

---

## [Author Response · Author response to Decision Letter 0]

24 Oct 2022

Thank you very much for reviewing our manuscript. Please see below for our responses to the reviewers. 

Reviewer #1 Comments:

1) The topic is relevant, interesting. However, Line 80- 82 describes previous research conducted on the topic. This indicates the sole novelty of this work being a focus on the use of algorithms in decision making, within the context of childhood asthma. if so, would it have been more worthwhile to gather insights from a larger sample on this topic?

Our response: 

Thank you for your positive feedback.

The reviewer is right that the novelty is algorithms in childhood asthma. The research reported was a qualitative sub-study within a clinical trial. The sample size was limited by the trial study design - it was not possible to resource the recruitment of a larger sample. Nevertheless, the data that we obtained from the interviews was both in-depth and extensive. We reached data saturation for both samples (families and clinical staff). We recommend that teams consider our findings and whether it is necessary for them to pursue small/in-depth vs. larger surveys on related research to develop knowledge on this topic further. This suggestion has been included within the discussion section of the paper (lines 517-520 – tracked changes).

2) Line 73: Additional details relevant to the cited prevalence rate would be useful such as location. Additional information on the global burden of asthma or at least, in the UK would be useful additions here as well.

Our response: 

Thank you for this suggestion. 

We have now included more information about the number of people diagnosed with asthma in the UK and clarified the prevalence of asthma in the paediatric population worldwide. This is highlighted in the introduction section (lines 67-70 – tracked changes). 

3) Line 117: The study design is described as a process evaluation, while the title indicates this is a qualitative study. Further, the title as well as early sections which preceded the Methods section point towards a study focusing on the lived experiences of staff and families in this context. The methods section does not sufficiently describe the focus on lived experience, which is essentially the methodology underpinning this work. It also seems quite irrelevant to explore perspectives related to recruitment and randomisation in this context. Please consider revisiting the framing of the methods section, consider the methodological merits of phenomenology in this research context and either changing the title to indicate this was a process evaluation or reframing the methods section to indicate being more strongly guided by qualitative methodology.

Our response: 

Thank you for pointing this out. We have amended the paper to clarify that this was a process evaluation. 

The process evaluation initially set out to explore experiences and ascertain acceptability of the trial intervention, and to solicit in depth feedback on taking part in this trial from the perspectives of both patients and trial staff. The process evaluation used qualitative interviews to collect data, but it was not designed to be an in-depth qualitative study with an underpinning philosophy used to guide the conduct of it. We nevertheless elected to report the findings in this paper as additional insights that emerged around the use of technology in the management of childhood asthma. We have clarified this within the Design sub-section of the Methods (lines 106-112 – tracked changes). More sophisticated analyses was therefore conducted to examine these additional insights – outlined in the findings section.

4) Line 135-140 and Line 151-156 are repetitive. Please consider a creative way to present this section to address this.

Our response:

Thank you for pointing this out. We have now changed the wording of these sentences. 

5) Line 143 states "set out to interview 20 families". However, the final sample size has not been described.

Our response: 

The final sample size is described on page 10. We interviewed 6 mother-child pairs.

6) Line 169: Identification of themes post listening to the audio recording is not a commonly used method. Perhaps use a reference to indicate it has been used by others. Also, it would be useful to explain why member checks were not conducted (e.g. lack of resourcing). Especially, as Line 176 notes findings were reviewed by the clinical teams. Additionally, this is not traditionally an approach undertaken to ensure the findings authentically represent the user experience, which is the purpose of member checks.

Our response: 

Thank you for this suggestion. We have included a reference [22] as suggested.

We recognise the merit of conducting member checks. However, we would like to re-iterate that this was a process evaluation which was designed to solicit in depth feedback on taking part in the trial from the perspectives of both patients and trial staff. Findings were reviewed by clinical teams in order to facilitate the conduct of the trial. It would have been difficult to complete member checks in this context – data collection was undertaken in the first year of the pandemic, when clinical staff time was even more costly. We may have conducted member checks for the data-set arising from the family interviews, but this study was designed as a process evaluation as opposed to an in-depth qualitative project and therefore member checks were not done. 

8) Line 191-193: This seems a poor rationale for the sample size of families recruited into the study, given the focus is on medication adherence, and therefore in particular the experiences of the users.

Our response: 

The increased sample size of clinical staff, compared to families, arose namely due to feedback which suggested that there were multiple occasions when clinical staff deviated from the treatment recommendations provided by the algorithm. We therefore shifted our focus to establish the reasons behind the deviations. The process for sample selection (families) is described in lines 147-156 (tracked changes). 

We would like to highlight that it was difficult to recruit families – we cannot say with certainty that this was related to the pandemic but do acknowledge that it could have been. Families were approached on 4 separate occasions – firstly by research nurses at selected sites, and thereafter on 3 separate occasions by the qualitative researchers. Families who initially expressed an interest in taking part did not reply to further invitations. 

9) If there were 17 families who expressed an interest in participating, did the participant information sheet state that recruitment would be discontinued once saturation had been reached? Did the research team notify the remaining prospective participants that they would not be interviewed as data saturation had been reached? I feel these are important aspects to be addressed in the study design and description.

Our response: 

The participant information sheet did not state that recruitment would be discontinued once saturation had been reached. The remaining (prospective) participants were not notified that they would not be interviewed because they did not respond to follow-up invitations (telephone calls). We have included more information related to the procedure used to follow-up potential participants in an Appendix (S1 Appendix). This is also referenced within the manuscript. 

10) The results section is interesting and has been presented well.

Our response:

Thank you for this positive feedback. 

11) Line 473-474: is this a novel findings or replicates the study findings referred to in text as (31)?

Our response: 

This is a novel finding - we have now clarified within the manuscript that citation 31 is a research note (opinion piece) which discusses interpretations of surveillance and digital technology in healthcare, as opposed to a research study. 

12) Line 478: The strength of this claim will benefit from additional references, if they can be sourced.

Our response: 

Thank you for highlighting this. We have now included an additional reference [33] to support this claim. 

13) Line 505-506: This claim is not entirely accurate as the methods section noted the first researcher developed themes post listening to audio recordings and therefore could not have undertaken coding of the data.

Our response: 

We would like to clarify that listening to the audio recordings was part of the process of coding data and developing themes to describe key issues highlighted. This has been clarified in lines 177-178 (tracked changes). Both researchers coded the data and developed themes. 

14) I wondered if the data revealed user and staff experiences related to using this technology during a pandemic. It would be an interesting angle to explore if there is existing data on this within the interviews.

Our response: 

We agree that this would have been an interesting angle to explore. We did not collect any data related to the use of technology during the pandemic. 

15) Please review the reference list to ensure formatting is consistent (e.g. journal name is presented in upper and lower case)

Our response:

Thank you for pointing this out. This has now been rectified. 

16) Overall, the findings are relevant, and interesting. However, I would urge the authors to consider the concerns noted above and the overall strength and generalizability of these findings given the small sample size of parents interviewed. Perhaps, it is worthwhile considering the collection of additional data from families who meet the same inclusion criteria and increasing the overall sample size of the research undertaken. 

Our response: 

Thank you for your review. Please refer to our previous responses to comments 1 and 8 above. 

Reviewer #2 Comments 

Reviewer: The paper is logically structured and well-written. The authors could expand the introduction including additional information on the prevalence of asthma.

Our response: 

Thank you for the positive feedback. 

We have now included this information in the introduction section (lines 67-70 – tracked changes). 

Reviewer: In the method, it is recommended to include final sample sizes for staff and families and add justification for the sample size selections. 

Our response:

Please refer to our responses to comments 5 and 8 from Reviewer 1. Lines 125-163 (tracked changes) in the revised paper describe the processes for selecting the samples. The final sample size is described in line 193 (clinical staff) and line 209 (families). 

Reviewer: It is recommended to justify through references the use of audio recordings. 

Our response: 

Please refer to our response to comment 6 from Reviewer 1.

Reviewer: The results are presented well. The authors could strengthen the claims with additional references.

Our response: 

Thank you. We have presented additional references in the discussion section. 

Reviewer: It would be interesting to expand the finding related to the use of technology in pandemics if applicable. 

Our response: 

We agree that this would have been an interesting angle to explore, this has also been picked up by Reviewer 1. Please refer to our response to comment 14.

---

## [Decision Letter · Decision Letter 1]

7 Dec 2022

PONE-D-22-10382R1

A qualitative process evaluation within a clinical trial that used healthcare technologies for children with asthma – insights and implications

PLOS ONE

Dear Dr. Lawrie,

Thank you for submitting your manuscript to PLOS ONE. After careful consideration, we feel that it has merit but does not fully meet PLOS ONE’s publication criteria as it currently stands. Therefore, we invite you to submit a revised version of the manuscript that addresses the points raised during the review process.

Thanks for the revisions, please address the final comment from R2: "the final sampling size with justification for selecting this size, in the section "Sampling and recruitment". 

We look forward to receiving your revised manuscript.

Kind regards,

Daswin De Silva

Academic Editor

PLOS ONE

Journal Requirements:

Reviewers' comments:

Reviewer's Responses to Questions

Comments to the Author

1. If the authors have adequately addressed your comments raised in a previous round of review and you feel that this manuscript is now acceptable for publication, you may indicate that here to bypass the “Comments to the Author” section, enter your conflict of interest statement in the “Confidential to Editor” section, and submit your "Accept" recommendation.

Reviewer #1: All comments have been addressed

Reviewer #2: (No Response)

2. Is the manuscript technically sound, and do the data support the conclusions?

Reviewer #1: (No Response)

Reviewer #2: (No Response)

3. Has the statistical analysis been performed appropriately and rigorously?

Reviewer #1: (No Response)

Reviewer #2: N/A

4. Have the authors made all data underlying the findings in their manuscript fully available?

Reviewer #1: (No Response)

Reviewer #2: Yes

5. Is the manuscript presented in an intelligible fashion and written in standard English?

Reviewer #1: (No Response)

Reviewer #2: Yes

6. Review Comments to the Author

Reviewer #1: (No Response)

Reviewer #2: Thank you for your comments and revisions,

Regarding the review comment about the final sample size.

Authors should update the final sampling size with justification in the section "Sampling and recruitment". Without this, it is confusing and hard to follow from the reader's perspective.

7. PLOS authors have the option to publish the peer review history of their article (what does this mean?). If published, this will include your full peer review and any attached files.

Do you want your identity to be public for this peer review? For information about this choice, including consent withdrawal, please see our Privacy Policy.

Reviewer #1: No

Reviewer #2: No

---

## [Author Response · Author response to Decision Letter 1]

9 Dec 2022

Editor comment: 

Thanks for the revisions, please address the final comment from R2: "the final sampling size with justification for selecting this size, in the section "Sampling and recruitment". 

Authors' response: 

Thank you for the positive feedback on our manuscript. We have now included the final sample size in the “Sampling and recruitment” section of the methods (line 135, tracked changes – staff sample and line 155, tracked changes – family sample). The justifications for sampling size/method/criteria is also highlighted in the same section (lines 117-123, tracked changes – staff sample, and lines 154-157, tracked changes – family sample). This information was originally presented in the findings section.

We have changed the order of the references to accommodate for revised changes (as specified above) that included references.

---

## [Editor Report · Decision Letter 2]

20 Dec 2022

A qualitative process evaluation within a clinical trial that used healthcare technologies for children with asthma – insights and implications

PONE-D-22-10382R2

Dear Dr. Lawrie,

We’re pleased to inform you that your manuscript has been judged scientifically suitable for publication and will be formally accepted for publication once it meets all outstanding technical requirements.

Kind regards,

Daswin De Silva

Academic Editor

PLOS ONE
---

## [Editor Report · Acceptance letter]

26 Dec 2022

PONE-D-22-10382R2 

A qualitative process evaluation within a clinical trial that used healthcare technologies for children with asthma – insights and implications 

Dear Dr. Lawrie:

I'm pleased to inform you that your manuscript has been deemed suitable for publication in PLOS ONE. Congratulations! Your manuscript is now with our production department. 

Kind regards, 

on behalf of

Dr. Daswin De Silva 

Academic Editor

PLOS ONE